# A Comprehensive Review of Complications and New Findings Associated with Anorexia Nervosa

**DOI:** 10.3390/jcm10122555

**Published:** 2021-06-09

**Authors:** Leah Puckett, Daniela Grayeb, Vishnupriya Khatri, Kamila Cass, Philip Mehler

**Affiliations:** 1ACUTE Center for Eating Disorders, Denver, CO 80204, USA; leah.puckett@dhha.org (L.P.); Daniela.Grayeb@dhha.org (D.G.); Vishnupriya.Khatri@dhha.org (V.K.); Kamila.Cass@dhha.org (K.C.); 2Department of Medicine, School of Medicine, University of Colorado, Aurora, CO 80045, USA; 3Eating Recovery Center, Denver, CO 80230, USA

**Keywords:** eating disorders, anorexia nervosa medical complications, medical findings in anorexia nervosa

## Abstract

Anorexia nervosa is a complex and deadly psychiatric disorder. It is characterized by a significant degree of both co-occurring psychiatric diseases and widespread physiological changes which affect nearly every organ system. It is important for clinicians to be aware of the varied consequences of this disorder. Given the high rate of mortality due to AN, there is a need for early recognition so that patients can be referred for appropriate medical and psychiatric care early in the course of the disorder. In this study, we present a comprehensive review of the recent literature describing medical findings commonly encountered in patients with AN. The varied and overlapping complications of AN affect pregnancy, psychological well-being, as well as bone, endocrine, gastrointestinal, cardiovascular, and pulmonary systems.

## 1. Introduction

Anorexia nervosa (AN) is a complex psychiatric disorder with a high rate of mortality and a relatively low rate of remission [1]. Using DSM-V criteria, the lifetime prevalence of AN in females is estimated to be as high as 4% [2]. The lifetime prevalence in males has been estimated to be between 0.1% and 0.3%, although this is likely an underestimate [3]. AN has a high rate of psychiatric comorbidities including strong associations with mood and anxiety disorders, personality disorders, self-harm and suicidality, as well as substance use disorders [4,5]. In addition, AN leads to widespread medical complications across virtually all organ systems, which contributes to it being one of the highest causes of mortality among psychiatric disorders [6]. Complications worsen with a lower body mass index (BMI), which indicates a greater severity of disease [7]. Medical complications have been related to a plethora of complex physiological changes that lead to decreased energy expenditure and include cardiac, bone, obstetric, and gynecological changes, as well as endocrine, gastrointestinal, hematological, electrolyte imbalance, and skin changes [4].

## 2. Etiopathogenesis

The etiopathogenesis of AN is still unknown. Although progress has been made in identifying genetic, developmental, psychological, and neurobiological factors that play a role in the development of the disorder [8], it is evident that, similar to all psychiatric disorders, its origins are multifactorial. AN is familial, with heritability estimates for AN ranging from 0.41 to 0.74 [9,10], and higher heritability estimates have been found when more stringent definitions of anorexia were applied [11]. Significant genetic correlations have been found among AN and various anthropometric and metabolic traits, including negative correlations with BMI, fasting insulin, and fasting glucose [12], and therefore metabolic factors, including the gut microbiome, have been considered for understanding the etiology of AN [13,14]. Developmentally, cesarean section, multiple births, low gestational age, congenital malformations of the mouth or digestive system, and older parental age are associated with increased risk of AN [15,16]. Temperamental characteristics associated with lifetime prevalence of AN include perfectionism, obsessionality, reward dependence, negative affectivity, and neuroticism [17,18,19]. Neurobiologically, findings of reward processing and reward learning abnormalities indicate reward circuit dysfunction in AN [20,21]. Neuropsychologically, impairments such as set-shifting difficulties and poor central coherence may be related to the pathophysiology of the disorder [22]. Dieting is the most common precipitating factor for the development of AN, with severe dieting being especially implicated in triggering the onset of AN [23,24].

## 3. Psychiatric Treatment

There are currently no FDA approved medications for the treatment of AN [25]. The most established psychological treatment for adolescents with AN is family-based therapy (FBT) [26]. Randomized controlled trials have shown adolescent FBT to be superior to individual therapy for supporting weight restoration in adolescents. Currently, there are no therapy modalities that have achieved empirical support for adults with AN and no evidence that a particular mode of therapy is more efficacious than another mode of therapy [27]. Novel approaches are being explored, including treatments focused on neurocognitive factors found to be impaired in individuals with AN, including problems with set-shifting and central coherence [28]. Treating patients at the appropriate level of care is important. In severe cases, initial medical stabilization, followed by inpatient and residential treatment, is recommended [29]. Severe AN is defined by a BMI of <15 kg/m^2^. For adolescents, inpatient treatment of AN has been shown to be highly effective, with sustained increases in body weight and decreases in eating disorder symptoms found at one-year follow-up [30]. Due to a lack of insight into the severity of the disease among patients, involuntary treatment may be indicated [31].

## 4. Medical Complications

### 4.1. Endocrine System Complications

Anorexia nervosa (AN) is characterized by alterations in multiple neuroendocrine axes and peptides that signal or regulate energy intake. Some of the endocrine and metabolic abnormalities in patients with AN represent physiological adaptive responses to chronic starvation that are reversible after weight restoration. However, other abnormalities may play a role in disease pathophysiology and sustained neuropsychiatric symptoms. Hormonal changes include growth hormone (GH) resistance with low insulin-like growth factor-1 (IGF-1) levels, hypothalamic hypogonadism, hypercortisolemia, and changes in appetite-regulating hormones such as leptin, ghrelin, peptide YY, and possibly adiponectin [32]. Changes are seen in the hypothalamic-pituitary axis (HPA) in patients with AN which affect both anterior and posterior derived pituitary hormones.

#### 4.1.1. Hypothalamic-Pituitary-Adrenal Axis 

The hypothalamic-pituitary-adrenal axis (HPAA) is in a chronically stimulated state in at least one-third of patients with AN [33]. Elevated baseline cortisol levels are likely observed during extreme caloric restriction in both healthy individuals and patients with AN. However, reversibility of HPAA dysregulation, even in recovered patients with AN, is not always observed [34]. Cortisol stimulates gluconeogenesis, and cortisol levels in patients with AN have been shown to be inversely correlated to fasting glucose [35]. The degree of hypercortisolemia also correlates inversely with BMI and fat mass [36]. Furthermore, it is associated with the severity of bone loss and depression in patients with AN [37].

#### 4.1.2. Hypothalamic-Pituitary-Thyroid Axis 

Severe weight loss in patients with AN is characterized by the nonthyroidal illness syndrome (also known as euthyroid sick syndrome) seen in patients with systemic illnesses, including chronic starvation. In women with AN, levels of total triiodothyronine (T3) are low. The reverse T3 level is elevated due to increased peripheral deiodination of thyroxine (T4) to reverse T3. Free T4 levels vary from normal to low normal, and TSH levels vary from normal to low normal [38]. These changes are an adaptive response to a decrease in metabolic rate and energy expenditure, and therefore require no treatment other than weight recovery.

#### 4.1.3. Hypothalamic-Pituitary-Gonadal Axis 

Studies of adult women with AN show immature, low amplitude luteinizing hormone (LH) pulses. The altered LH pulsatility manifests as hypothalamic amenorrhea. LH secretory patterns may revert to prepubertal levels of low, nonpulsatile secretion or to a pubertal pattern of entrainment of LH secretion to the sleep cycle. LH and FSH responses to GnRH appear normal, indicating that the reduced pulse frequency is not secondary to pituitary changes but presumably because of changes in the GnRH pulse generator. The gonadotropin secretion is stimulated by both estradiol and leptin, which are both decreased in patients with AN [39].

#### 4.1.4. Growth-Hormone/Insulin-Like Growth Factor-1 Axis 

Anorexia nervosa (AN) is characterized by a nutritionally acquired growth hormone (GH) resistance leading to low concentrations of insulin-like growth factor-1 (IGF-1), which is another essential determinant of reduced bone mineral density. High levels of GH in individuals with AN have a gluconeogenic role in maintaining euglycemia, mediated through increased lipolysis. Weight gain leads to a normalization of GH secretion and GH-binding protein concentrations, which are consistent with adaptation to the individual’s nutritional status [40].

#### 4.1.5. Hypothalamic Neuropeptides 

Hypothalamic neuropeptides such as nesfatin-1, phoenixin, spexin, and kisspeptin affect energy homeostasis and eating behaviors and may also be involved in the pathogenesis of anxiety related to eating disorders [41].

#### 4.1.6. Vasopressin (or Antidiuretic Hormone, ADH) 

Hyponatremia is very common in patients with AN and may lead to complications such as altered levels of consciousness and seizures. Purging behaviors, as seen in the binge eating-purging subtype, are the most common cause, although the syndrome of inappropriate antidiuretic hormone secretion (SIADH) may also be implicated. Central diabetes insipidus (DI), resulting in hypernatremia, has also been reported in patients with AN due to defective vasopressin secretion [42]; DI can also manifest during the refeeding period as a disorder of osmoregulation between serum osmolality and plasma vasopressin levels [43].

#### 4.1.7. Oxytocin 

Basal levels of oxytocin, a neuropeptide normally produced in the hypothalamus and released by the posterior pituitary, are decreased in patients with AN. Oxytocin plays a role in social bonding, modulation of anxiety and depressive symptoms, and bone metabolism. It is speculated that low oxytocin levels may contribute to alexithymia in women with anorexia nervosa [44].

#### 4.1.8. Appetite-Regulating Hormones 

Gastrointestinal hormones are normal signals to modulate appetite and maintain energy homeostasis. Ghrelin is an orexigenic hormone predominantly produced in the stomach and is thought to act upon the HPA, affecting the secretion of gonadotropin-releasing hormone (GnRH), adrenocorticotropic hormone (ACTH), growth hormone (GH), follicle-stimulating hormone (FSH), and LH. Plasma ghrelin levels are elevated in both AN-restricting (AN-R) and AN-binge eating/purging (AN-BP) subtypes. The increase in a hormone with orexigenic action in patients with AN is paradoxically caused by a central resistance to ghrelin. In patients with AN, the lack of a motivating feeding response to ghrelin, despite hunger sensation, supports an altered reward circuit that may contribute to the disease’s onset and maintenance [45]. Elevated ghrelin concentrations in patients with AN may be necessary to modulate glucose homeostasis, serving as an adaptive mechanism for maintaining normoglycemia in response to severe undernutrition. The secretion of ghrelin varies with insulin levels, suggesting a reciprocal regulation of these hormones [46].

Peptide YY (PYY) is an anorexigenic hormone released by the L cells of the distant intestine in response to caloric intake that acts at the hypothalamus to decrease appetite and food ingestion. PYY correlates inversely with BMI and fat mass. Levels of PYY are paradoxically elevated in patients with AN; this would not be adaptive to a state of chronic malnutrition but may play a role in the disorder’s pathogenesis contributing to decreased nutrient intake and disordered eating psychopathology [47,48].

Adipocytokines secreted by adipose tissue, such as adiponectin, leptin, and resistin, have significant roles in regulating energy metabolism and insulin sensitivity [49]. Leptin, a hormone mainly produced in white adipose tissues, is a valuable marker of long-term energy stores. In patients with AN, there is a marked fall in leptin concentration in proportion to fat mass [46]. A decrease in serum leptin levels, described in the literature, has been postulated to lead to both amenorrhea and restlessness similar to rat-specific semi-starvation-induced hyperactivity [50]. However, data concerning serum levels of the other adipocytokines are conflicting. Serum adiponectin has been found to be either increased, normal, or decreased in various studies in AN patients [51,52]. Likewise, serum plasma resistin levels have been found to be either decreased or normal [53,54]. These discrepancies suggest that confounding factors, such as eating behaviors and physical activity level, could interfere with the results and further studies are needed.

#### 4.1.9. Glucose Metabolism

In general, fasting levels of glucose are lower in patients with AN than in healthy controls. Hypoglycemia has been described as one of the causes of sudden death in patients with AN. Liver injury caused by autophagia, hypotension, ischemia, in addition to marked glycogen depletion, seem to be the causes of hypoglycemia. To counteract the effects of hypoglycemia, secretion of both GH and cortisol induce insulin resistance, and thus increase glycemia. Gastrointestinal hormones such as glucagonlike peptide 1 (GLP-1) and amylin stimulate insulin release from pancreatic cells. The plasma levels of GLP-1 and amylin are both found to be low in patients with AN, which seems to be an appropriate adaptive response to starvation in order to prevent further hypoglycemia [42].

### 4.2. Refeeding Syndrome

The term refeeding syndrome (RS) has been used to describe the adverse consequences that can occur in all malnourished patients during the early stages of nutrition repletion, whether the method of refeeding is oral, enteral, or parenteral. Patients with AN are at high risk for RS. Refeeding hypophosphatemia (RH) is the most common complication of nutritional restoration for patients with AN. The highest risk of hypophosphatemia seems to be in patients who weigh less than 70% of their ideal body weight or lose weight rapidly [55]. Other consequences of RS include acute thiamine deficiency resulting in Wernicke’s encephalopathy and Korsakoff syndrome, with the potential for permanent cognitive impairment, hypokalemia, hypomagnesemia, metabolic acidosis, or alkalosis, and fluid overload resulting in cardiac failure. The American Society for Parenteral and Enteral Nutrition (ASPEN) Parenteral Nutrition Safety Committee and the Clinical Practice Committee have developed consensus recommendations for identifying patients with or at risk for refeeding syndrome (RS) and for avoiding and managing the condition. RS diagnostic criteria can be stratified as follows: a decrease in any one, two, or three of serum phosphorus, potassium, or magnesium levels by 10–20% (mild), 20–30% (moderate), or >30% and/or organ dysfunction resulting from a decrease in any of these or due to thiamine deficiency (severe), occurring within 5 days of reintroduction of calories [56]. In patients with AN, the degree of malnourishment appears to correlate with the severity of RH [57].

The reintroduction of nutrition leads to a switch from fat to carbohydrate metabolism and an increase in insulin concentration. Insulin stimulates the movement of potassium, phosphate, and magnesium into cells leading to their depletion in the extracellular compartments. Reactivation of carbohydrate metabolism increases degradation of thiamine, a cofactor required for cellular enzymatic reactions in Krebs cycle. Deficiency in all these nutrients can then occur [58].

Recent studies have provided evidence to support a switch in current care practices for refeeding from a conservative approach to higher calorie refeeding; however, caution should still be applied for more severely malnourished, i.e., <70% average body weight, and/or chronically ill, adult patients [59].

### 4.3. Bone Health in Anorexia Nervosa

Of notable concern related to AN are the effects of malnutrition on bone health. Unfortunately, the deleterious effects on bone mineral density (BMD) are not completely reversible despite weight restoration. These effects include decreased height, chronic pain, and increased immediate and lifetime risk of fracture due to inadequate bone accrual and ongoing bone loss [60,61]. A recent cohort study of 344 female patients who were being treated for eating disorder found a significantly lower BMD at the L-spine (16%), femoral neck (18%), and total hip (23%) in patients with active AN. Patients in remission had lower BMD than healthy controls but higher than women with active AN at the L-spine and hip [62].

Bone mineral density is measured by dual-energy X-ray absorptiometry (DEXA). T scores reflect a comparison with peak bone mass of healthy adults, while Z scores reflect age and sex-matched comparisons. Generally, in younger patients (<50 years old), Z scores are used for reporting. Osteopenia is defined as a BMD between −1.5 and −2.5, while osteoporosis is defined as a BMD < −2.5 [63].

Males and females with AN are both at risk of bone disease; 80% of females with AN have BMD Z scores < 1, 44% of females have BMD Z scores < 2, and 32% of males with AN have BMD Z scores < 2 [64]. A previous longitudinal study demonstrated a decrease in BMD over time, with a large mean annual decline of 2.4% at the hip and 2.6% at the spine. While BMD stabilized over the first year after weight restoration and continued to increase over time, it failed to fully normalize [64]. A recent study that assessed long-term outcomes of bone health in females, at 5 and 10 years after initial diagnosis of anorexia nervosa, demonstrated decreased BMD at the femoral neck and arms and CT evidence of cortical thinning, despite BMI being normal [65].

Factors associated with an increased risk of low BMD include lower BMI, longer duration of illness and amenorrhea, decreased muscle mass, and lower serum vitamin D levels [61,66]. The risk appears to be increased primarily in patients with restrictive eating disorder patterns. One large community sample found that BN was not a risk factor for low BMD if not associated with restriction [67]. An additional risk factor for bone disease in patients with AN is adolescent age. An earlier age of onset of AN (<18 years) is associated with a greater decrease in BMD [64]; 39% of bone mineral content is accrued between ages 10–14 and 95% of peak bone mass is typically reached prior to age 19 in healthy females, a process that is impaired in patients with AN. Adolescent females as well as males demonstrate decreased markers of both bone formation and resorption [64]. Adult women with AN demonstrate increased bone resorption and decreased bone formation [64]. This “uncoupling” is indeed the reason for aggressive loss of BMD in patients with AN, notwithstanding their typically young age, in contrast to post-menopausal women with osteoporosis where the only issue is increased resorption.

Bone disease in AN is caused by a complex interplay of physiological changes that occur in the malnourished state in response to energy deprivation which are incompletely understood. Changes in function of the hypothalamic-pituitary-adrenal axis include decreased GNRH leading to decreased LH and FSH and impaired ovulatory function. This leads to decreased estrogen which results in increased bone resorption. Decreases in testosterone and dihydroepiandrosterone (DHEA) are also observed and can further lead to decreased estrogen. In males with AN, low testosterone contributes to decreased BMD [68] and indeed severe osteoporosis is also a major problem for these males. In addition, GH resistance and increases in CRH, ACTH, and cortisol lead to decreased BMD. Changes in hormones such as leptin which is stored in adipose tissue and alterations in gut hormones further contribute. Collectively, these changes are reflective of the energy-conserving state in AN [64].

Patients with AN demonstrate a lifetime increased risk of fracture in both males and females. Thirty percent of women with AN report a fracture at one or more sites over a lifetime [67]. Risk of fractures in females is observed at younger ages, across all sites, whereas risk of fractures in males with AN is seen after age of 40 with an increased rate of vertebral fractures specifically [64]. Cumulative incidence of any fracture persists 40 years after diagnosis [69].

Despite the evidence of bone disease as evidenced by both decreased BMD and increased fracture risk in patients with AN, it is unclear if the increased fracture risk seen in AN can be directly associated with BMD measurement. One study, which demonstrated 60% increased fracture risk in women with AN, found fractures occurred even in women with normal BMD. Another 18-month longitudinal study demonstrated that 12.5% of women with AN had asymptomatic vertebral fractures not associated with illness duration, severity of malnutrition, or BMD [64]. A recent study that demonstrated increased fracture risk in AN did not find a direct association with decreased BMD [62]. Authors cited potential limitations of DEXA for assessing BMD in patients with AN, including relationships between DEXA and fracture risk taken primarily from post-menopausal populations, altered body composition arising from cycles of weight-loss and weight-gain leading to inaccurate estimate of BMD, and difficulty estimating the cumulative effects of AN given multiple periods of disease relapse and remission [62]. Further studies need to be conducted to better elucidate how to utilize DEXA in evaluating and reducing fracture risk in patients with AN.

Given the interplay between amenorrhea and bone disease, there has been interest in hormonal treatment of both males and females with AN. Many studies have not demonstrated increased BMD in females with AN who used oral contraceptive pills (OCPs), and there has been concern regarding hormonal treatment in younger patients with AN due to the risk of premature closure of epiphyseal plates [64]. However, a recent cross-sectional study demonstrated possible benefits. Females with AN using OCPs were found to have improved BMD at whole body, lumbar spine, as well as femoral neck, hip, and radius. The benefits correlated with longer duration of treatment with OCPs and shorter delay in initiating OCP treatment after AN onset. Patients with lower BMI experienced the greatest benefit from use of OCPs [70]. Yet, most experts do not recommend OCPs to treat low BMD in patients with AN. Previously, physiologic estrogen has been shown to improve BMD at the spine and hip in adolescents with AN [71]. Another recent small exploratory study that investigated the use of transdermal estrogen in adult women with AN found increased spine BMD [72], and many experts have recommend transdermal estrogen for younger patients with AN and low BMD. Testosterone therapy may be beneficial for treating males with AN. In older men with osteoporosis, testosterone has been shown to increase BMD, as well as in younger men with hypogonadism. However, there is a lack of data on treatment of bone disease in males with AN [68].

The mainstay for treatment of bone disease in AN is nutrition and weight restoration. In addition, patients should be instructed to take an adequate amount of calcium and vitamin D to optimize bone health. Studies in patients with AN have demonstrated an association with calcium intake <600 mg/day and lower bone mineral densities, as well as a high prevalence of vitamin D deficiency and insufficiency [68]. Treatment of bone disease in AN with prescription medication must take into account the risks and benefits of the various potential options. Bisphosphonates, which work by inhibiting bone resorption, are one option. Alendronate has been shown to increase BMD in adolescents treated for one year; however, the improvement compared to controls was not statistically significant. A randomized controlled trial demonstrated improvement in spine and hip BMD in adult women with AN [68]. Possible adverse effects include congenital abnormalities in women of childbearing age, which are particularly concerning given the long half-life of bisphosphonates [68]. Other potential side effects are osteonecrosis of the jaw in patients undergoing extensive dental work, gastrointestinal side effects, and atypical femur fractures [63]. Another treatment option is teriparatide, a recombinant parathyroid hormone analog which improves bone formation. It is given as a daily injectable for at least 2 years which may be impractical for some women. Yet, it has demonstrated significant improvement in BMD (10.5% at the spine) in older women with AN [73]. Teriparatide is contraindicated in pregnancy and should be avoided in patients with Paget’s disease or unexplained alterations in alkaline phosphatase given a risk of development of osteosarcoma observed in rats [63,68]. Another option is denosumab, a human monoclonal antibody that inhibits osteoclasts and is an injection given every 6 months. This medication has been demonstrated to be effectiveness in post-menopausal women with osteoporosis but has not been studied in AN. BMD decreases quickly after cessation of denosumab, therefore, drug holidays are not recommended with its usage [63,68].

### 4.4. Gastrointestinal Complications in Anorexia Nervosa

Gastrointestinal (GI) symptoms are common in AN, with over 90% of patients reporting GI complaints. While the pathogenesis of GI complaints are attributed to sequelae of weight loss, there is also a significant functional component to many of the GI complaints by patients with AN [74,75].

A recent systematic literature review, from 1967 to 2019, categorized the most common subjective GI symptoms reported by patients with AN. Among them include constipation, nausea, abdominal pain, abdominal fullness, vomiting, heartburn, epigastric pain, decreased appetite, diarrhea, and dysphagia [76]. Kessler et al. found that the severity of GI symptoms was more significantly correlated with somatization than with eating disorder severity and did not correlate with BMI [77]. This supported a previous study by Boyd et al. that also reported an association with anxiety and neuroticism and severity of GI complaints in patients with AN [78]. A recent literature review (2000–2017) of the medical causes of food-related GI symptoms in eating disorders found that AN patients reported higher food-associated symptoms than average; however, the incidence of celiac disease, an immune-mediated disorder to gluten, was similar among patients with AN and the general population. On the basis of their comprehensive review, Kress et al. predicted that the prevalence of immunological or structural GI disorders was no higher in patients with AN than in the general population, and that GI complaints from patients with AN were more likely to be functional. They advised that further diagnostic workup for persistent symptoms was warranted only after control of purging and restrictive behaviors and weight restoration [75]. More studies in patients with AN are needed to assess the functional nature of GI complaints and to further evaluate the prevalence of food allergies and intolerances.

Another interesting area of research is the role of intestinal microbiota, i.e., living organisms including prokaryotes, eukaryotes, archaea, and viruses, on the gut-brain axis [79]. Hata et al. found that transplanting the gut microbiome from patients with AN into germ-free mice resulted in poor weight gain, decreased appetite, decreased food efficiency, increased anxiety-related and compulsive behaviors, and decreased serotonin levels as compared with healthy mice. Compulsive behavior improved after administration of *Bacteroides vulgatus* (a genus found to be low in AN patients), however, it did not impact weight gain [80]. Studies in patients with AN have shown a significant reduction in *Roseburia* species, which was associated with a decreased level of butyrate, i.e., a short-chain fatty acid, that correlated with anxiety and depression disorders. A reduction in *Roseburia inulinivorans* was correlated with lower insulin levels, which may help patients with AN to preserve euglycemia. Additionally, higher levels of *Enterobacteriacee* and *Methanobrevibacter smithii*, an Archaeon capable of extracting more calories from food through the transformation of hydrogen to methane, have been identified in patients with AN as compared with healthy controls. There is experimental evidence that intestinal methane production is linked to slower intestinal transit, predisposing to constipation [81]. One study showed that, while overall microbial richness increased after weight gain, gut dysbiosis, short chain fatty acid profiles, and GI complaints remained persistent three months after weight gain [74]. Whether dysbiosis is the cause or effect of AN remains to be seen. Furthermore, fecal microbiota transplant is a promising area of research that might provide treatment and shed more light into this complex system [74,81].

While many of the GI complaints in patients with AN may be functional, there are certain GI complaints that require further investigation. Many case reports have documented superior mesenteric artery (SMA) syndrome (more commonly in AN-R) [76,82]; acute gastric dilation with subsequent gastric wall ischemia, necrosis, and perforation (more commonly in AN-BP); and duodenal dilation (more commonly in AN-R) [76], which further exacerbate GI symptoms and should remain in the differential diagnosis of abdominal pain in patients with AN.

In addition, there are numerous abnormalities in the liver that occur in AN. Starvation-induced autophagy and apoptosis is the proposed mechanism for liver dysfunction in AN, supported by liver biopsies in patients with AN with aminotransferase > 1900 IU/mL (typically with ALT > AST) [83]. Independent of body weight, the incidence of hypoglycemia was four to five times higher with liver dysfunction, and hypophosphatemia occurred twice as often with elevated liver function tests in patients with AN [84]. Severe, acute liver failure in patients with AN is reversible with weight restoration and rarely requires testing for secondary causes of liver dysfunction. An increase in aminotransferases also occurs with refeeding, likely due to excessive hepatic fat and glucose deposition, and can be managed with a reduction in daily caloric amount [83].

### 4.5. Cardiovascular Complications in Anorexia Nervosa

Cardiovascular complications have long been implicated in sudden death of patients with AN, which, as previously stated, remains to be one of the deadliest psychiatric disorders [85]. Research advances over the past few years have progressed our understanding of the static and dynamic cardiac changes in patients with AN.

Studies of cardiac hemodynamics continue to demonstrate the near universal finding of bradycardia, as well as decreased heart rate variability (HRV) in patients with AN [85,86]. Bradycardia is reversed with weight restoration, and permanent pacemaker placement is unwarranted and causes procedural complications [85,86]. As Cotter et al. recommended, society guidelines should be updated to include AN within reversible causes of bradycardia to avoid unnecessary permanent pacemaker placement [86]. A meta-analysis demonstrated a small increase in resting vagal tone, measured indirectly through HRV [87]. A decreased HRV has also been seen in patients with AN who underwent continuous cardiac monitoring for a year [85]. Co-morbid depression further decreased the HRV in patients with AN [88]. Decreased HRV has been a known independent predictor of mortality in post-myocardial infarction patients for decades [89]. However, more studies are needed to understand the clinical implication of decreased HRV in patients with AN.

Our understanding of cardiac structure and function in patients with AN continues to progress with comprehensive, modern, and dynamic echocardiographic evaluations and more recently cardiac MRI (CMRI) studies. A retrospective study evaluated Doppler echocardiograms in adult patients with AN with an average BMI of 12 (N = 124) [90]. All AN patients had subclinical cardiac impairments and 15% had reduced ejection fraction (EF). Interestingly, AN-BP and hypertransaminasemia were independently associated with a reduced EF. Consideration for echocardiogram should be made in patients with AN (especially AN-BP) with hypertransaminasemia to prevent complications of heart failure with refeeding [90]. Doppler echocardiography used to assess aortic stiffness by pulse wave velocity in adolescent AN patients found increased arterial stiffness [91]. Whether this portends a higher risk of cardiovascular disease, or if it is reversed with weight restoration, remains to be seen. A meta-analysis of echocardiographic findings in patients with AN revealed a reduction in left ventricular mass, a reduction in cardiac output, diastolic dysfunction, and an association with mitral valve prolapse and pericardial effusion, with a trend toward improvement with weight restoration [92]. Left ventricular mass was increased in hyperactive patients with AN as compared with non-hyperactive patients with AN but was still significantly lower than in the healthy controls [93]. Dynamic changes in cardiac function during exercise have been evaluated using stress echocardiography [94]. Adolescent patients with AN had reduced exercise duration, which was independently associated with BMI, and reduced peak cardiovascular indices as compared with healthy controls, but a normal pattern of cardiovascular response with progressive exercise [94]. In a study of 40 patients with AN (average BMI 15), 23% of the patients showed evidence of myocardial fibrosis on CMRI as evidenced by late gadolinium enhancement (LGE) as compared with 0% in the control group [95]. Myocardial fibrosis has been positively correlated to sudden death in the general population, likely due to propensity for arrhythmia formation [95]. A more recent study did not find evidence of LGE on CMRI in patients with AN. However, the majority of AN patients had recovered weight at the time of CMRI in the study [96]. More CMRI studies in patients with AN are warranted to better understand the prevalence of myocardial fibrosis and the clinical significance.

Previous studies have reported conflicting results on whether QTc prolongation is inherent in AN [97,98,99,100]. Recent studies, however, have reported consistent findings of normal QTc in patients with AN, including a meta-analysis of 964 patients and a cohort study of 1026 patients, using a variety of QTc correcting formulas [97,98]. Furthermore, another study found no correlation between QTc > 440 ms and increased risk of cardiac events or all-cause mortality in patients with AN [99]. QTc prolongation, when present in patients with AN, therefore, should be considered in the context of extrinsic factors such as hypokalemia and/or known QT prolonging medications rather than being summarily attributed to AN [97]. A dynamic QTc evaluation during exercise showed a longer QTc in patients with AN than healthy controls during peak exercise, returning to normal at rest [100]. More studies are needed to validate this finding and control for QTc prolonging medications. As the focus shifts from QTc prolongation as a cause of sudden death of patients with AN, a year-long insertable cardiac monitor study revealed bradyarrhythmia, particularly sinus pauses, to be more common and perhaps more relevant for understanding the etiology of sudden cardiac death than ventricular tachyarrhythmias, although more studies are needed [85].

### 4.6. Pulmonary Complications in Anorexia Nervosa

Anorexia nervosa can lead to weakness and wasting of respiratory muscles, dyspnea, reduced aerobic and pulmonary capacity, lung parenchyma alterations, and life-threatening consequences [101]. In recent studies, cases of spontaneous pneumothorax and pneumomediastinum related to starvation have been reported [102,103,104]. Different mechanisms for this have been proposed. One proposed mechanism is a traumatic mechanism with vomiting-induced esophageal rupture (Boerhaave syndrome), as has been described in cases of AN. Another proposed mechanism is the Macklin effect, where alveolar rupture may occur due to increased intra-alveolar pressure and low perivascular pressure [105,106]. Air from ruptured alveoli may migrate into the mediastinum, pericardium, peritoneal, and retroperitoneal cavities; subcutaneous tissues; and the epidural space [107].

Bullae, bronchiectasis, and other structural changes of the lungs may occur as severe complications of AN [108]. Emphysema-like changes have also been reported in the lungs of chronically malnourished patients because of anorexia nervosa [109].

Lung infections due to opportunistic organisms and usually nonpathogenic mycobacteria have been described in patients with AN and no history of a preexisting pulmonary disease [110,111,112]. However, gastroesophageal disease and other causes of chronic aspiration (self-induced vomiting) may predispose patients with AN to develop bronchiectasis, and thus increase their risk for non-tuberculous mycobacterial infections [113].

### 4.7. Hematologic Complications in Anorexia Nervosa

Gelatinous marrow transformation occurs as malnutrition worsens. Specifically, serous fat atrophy in the bone marrow, and normal marrow fat is replaced by a thick mucopolysaccharide substance that impedes the egress of precursor cells from the bone marrow [114,115]. This leads to trilinear hypoplasia with leukemia, anemia, and thrombocytopenia detected in that order of decreasing frequency [116].

Interestingly, despite frank neutropenia, patients with AN do not appear to be at an increased risk of infection, and thus neutropenic precautions are not needed. Similarly, the use of expensive growth factors is not indicated because the marrow reconstitutes quickly with nutritional rehabilitation. Anemia in patients with AN is typically normocytic, but when the red blood cells indices are abnormal, it is typically macrocytic, although vitamin B₁₂ and folate levels are not low [117]. Microcytic anemia is rare and requires additional evaluation.

### 4.8. Obstetric/Gynecologic Complications in Anorexia Nervosa

Amenorrhea occurs in up to 84% of females with AN, related to complex physiological hormonal changes including the aforementioned drop in serum levels of leptin and a decrease in gonadotropin-releasing hormone which leads to decreased LH, FSH, and anovulation [118,119]. This has important consequences on fertility in women with AN, making it more difficult to conceive and increases the likelihood of miscarriage and pregnancy related complications such as preterm birth, cesarean section, microcephaly, small for gestational age, and perinatal mortality if conception does occur. These complications persist in women with a history of AN despite recovery, albeit with overall improvement in fertility potential [118,120]. Thus, women with AN need a higher level of surveillance during pregnancy and delivery [120].

Despite overall difficulties with fertility in patients with AN, adolescents with AN have a two-fold increase in unplanned pregnancy. A recent review found a lack of evidence for a negative impact on BMD in adolescents and young women with AN treated with combined hormonal contraceptives over a 12–18 month period [119]. In addition, there may be a protective effect of prolonged usage of OCPs [119]; however, this remains controversial. Given the high risk of unplanned pregnancies, contraception is an important issue to consider.

## 5. Conclusions

In summary, the complications of AN are varied and complex, with a significant overlap in pathophysiological mechanisms. The devastating effects of malnutrition perpetuate the eating disorder given exacerbation of physical and psychological discomfort during malnutrition and refeeding. Awareness of current and real medical complications that occur in patients with severe AN can help both clinicians and families to recognize the complexity of the illness and underscores the need for informed care. There is significant emerging research about the widespread medical impact of AN, the understanding of which helps providers to best serve patients who suffer from this disorder.

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
