# Peer review of "A Comprehensive Review of Complications and New Findings Associated with Anorexia Nervosa"

_jcm, 2021, doi:10.3390/jcm10122555_

Round 1

Reviewer 1 Report

This work is well organized and provides an overview of the major complications of AN. 

Author Response

Reviewer 1: Thank you for the affirmation of the value of this paper.

Reviewer 2 Report

The title says “ A comprehensive review..” however for this strong statement to be valid a much more thorough and complete walk through of the complications and in particular new findings in AN needs in my opinion to be done in order to accept this review. I would be happy to review a new version if you decide to put in more efforts. I have added some major and minor comments below. These should not be seen as a complete list of things that needs to be addressed, but as illustrative examples. Thus, I suggest you to work through the whole manuscript with the comments below in mind.

Comments:

The whole section on etiopathogenisis needs to be expanded.

 As it is now a bunch of fragmented factors are mentioned and no thorough review of the factors involved in the ethiology of AN seems to have been done.

r. 47-49. This statement needs to be backed up by many more citations, for each of the factors mentioned. In particular, teasing in childhood..

r. 53-54 please clarify what you mean with “the reinforcing nature of physical and interpersonal effects”

r. 61-65 “Randomized controlled trials have shown FBT to be superior to individual therapy in supporting weight restoration. Currently, there are no therapy modalities that have achieved empirical support for adults with anorexia and no evidence that a particular mode of therapy is more efficacious than another [14].” First you say that FBT is superior and then that there is no empirical support. Please clarify.

r. 67 Neurocognitive factors such as? Give examples

r. 78 You have already used the abbreviation AN, thus no need to spell it out once more. And then abbreviate on r.90… r.150.. r.229… r.488.

r. 82 singularly? Please rephrase.

r. 95-96… due to… due to..

r. 100-101 what do you mean with partially? Please explain.

r. 105-109 please rephrase sentence on T3/T4

r. 130-134 I cannot see what these hypothalamic neuropeptides has to do in the section on GH. Please explain or move.

r. 158 Please explain AN-R and AN-BP

r. 161. Make sure that it is correct that they have normal hunger sensation. How was that evaluated?

r. 170 PYY not PPY

r. 171 abbreviate PYY also here

r. 206, 215 abbreviate bone mineral density the first time you use it

r. 324-325 if not significant then there is no effect

r. 344 holidays?

r. 390. Please clarify what you mean with impair microbiota

r. 521 overlay?

Author contributions?

r. 837???

Go through the whole manuscript regarding the use of disease vs disorder

The same goes for Anorexia vs anorexia nervosa/AN. Anorexia literally only means loss of appetite and is accompanying several physiological conditions. Please always use anorexia nervosa/AN when referring to the psychiatric disorder.

Author Response

Reviewer 2: Thank you for your helpful comments.  Described below are the changes we have made.

The title says “A comprehensive review..” however for this strong statement to be valid a much more thorough and complete walk through of the complications and in particular new findings in AN needs in my opinion to be done in order to accept this review

Response: With regard to the comment regarding the title, this is a comprehensive review, and in fact we have further augmented its content.

The whole section on etiopathogenesis needs to be expanded. As it is now a bunch of fragmented factors are mentioned and no thorough review of the factors involved in the etiology of AN seems to have been done.

Response: In response to this suggestion, this section has been significantly modified (lines 43-68). However, the primary focus of this manuscript is complications, not etiopathogenesis.

  1. 47-49. This statement needs to be backed up by many more citations, for each of the factors mentioned. In particular, teasing in childhood..

 Response: This entire section has been re-worked and “teasing in childhood” omitted.

  1. 53-54 please clarify what you mean with “the reinforcing nature of physical and interpersonal effects”

Response: This statement has been removed.

  1. 61-65 “Randomized controlled trials have shown FBT to be superior to individual therapy in supporting weight restoration. Currently, there are no therapy modalities that have achieved empirical support for adults with anorexia and no evidence that a particular mode of therapy is more efficacious than another [14].” First you say that FBT is superior and then that there is no empirical support. Please clarify.

Response: This section has been clarified (lines 71-77) and there is not residual confusion.

  1. 67 Neurocognitive factors such as? Give examples

Response: Examples have now been given (lines 79-80)

  1. 78 You have already used the abbreviation AN, thus no need to spell it out once more. And then abbreviate on r.90… r.150.. r.229… r.488.

Response: This has been fixed and is now consistent throughout the manuscript.

  1. 82 singularly? Please rephrase.

Response: This has been rephrased.

  1. 95-96… due to… due to..

Response: Duplicate comment removed.

  1. 100-101 what do you mean with partially? Please explain.

Response: We addressed this (lines 113-114).

  1. 105-109 please rephrase sentence on T3/T4 

Response: We addressed this (see lines 118-122).

  1. 130-134 I cannot see what these hypothalamic neuropeptides has to do in the section on GH. Please explain or move.

Response: This has been moved to its own section (144-147).

  1. 158 Please explain AN-R and AN-BP

Response: These are acceptable abbreviations in ED literature. They have also been spelled out (lines 171-172).

  1. 161. Make sure that it is correct that they have normal hunger sensation. How was that evaluated?

Response: Upon further investigation we have determined they do not have normal hunger sensation.

  1. 170 PYY not PPY

 Response: Corrected (line 182)

  1. 171 abbreviate PYY also here

Response: Corrected (line 184).

  1. 206, 215 abbreviate bone mineral density the first time you use it

Response: Corrected (lines 261-262).

  1. 344 holidays?

Response: This is a term we are familiar with as used extensively in the literature, including a recent article in the NEJM.

  1. 390. Please clarify what you mean with impair microbiota

Response: This has been changed to “abnormal” microbiota (lines 451-453).

Author contributions?

Response: These have been described for each of the authors (lines 632-646).

Go through the whole manuscript regarding the use of disease vs disorder

Response: When describing levels of severity of AN, the term disorder is used. When we refer to eating disorder, we also use disorder, however we feel disease is a better fit when referring to medical issues/complications.

The same goes for Anorexia vs anorexia nervosa/AN. Anorexia literally only means loss of appetite and is accompanying several physiological conditions. Please always use anorexia nervosa/AN when referring to the psychiatric disorder.

Response: We have reviewed this and changed throughout the manuscript, in concert with your suggestions.

Reviewer 3 Report

Dear Authors,

this is a rather complete and very good review of current developments in the field of complications of anorexia nervosa. Congratulations. Because of the multiple endocrine changes, it is right to focus on endocrine system complications. The current controversial views on cardiac problems due to arrhythmias is on point. The lines of debate about the use of OCPs and transdermal estrogens are also well presented. I was particularly pleased with the section on pulmonary complications in AN; it is rare to read about this. I have only a few comments:

A reference to the refeeding syndrome is completely missing- it is certainly not necessary to present this elaborately, however, the refeeding syndrome refers to an artificial component of our clinical observation of complications in AN : It is a completely different situation whether I observe complications during the refeeding phase or during acute anorexia nervosa.

The structural changes of the organs due to AN, especially the histological changes in the bone marrow, liver, heart and kidney should be pointed out. This is clinically highly relevant , so fibrosis of the myocardium can lead to cardiomyopathy. Complications due to the frequent presence of laxative or diuretic abusus are missing, as is a brief section on renal problems, given the frequent need for dialysis in pat. with AN , quite justified.

I disagree with you that gastrointestinal problems are mostly functional in nature, as suggested by the extensive citation of Kress et al. There are a number of non-immunologically mediated intolerances, such as widespread lactose intolerance in pat. with AN. The inclusion of microbiome alterations in the review is good; worth considering is the seminal work of Hata et al. to include.

Finally, the limited literature on covid -19 and AN should be included and evaluated.

Author Response

Reviewer 3: Thank you for your favorable comments.

In reference to your comment RE: refeeding syndrome, we have now included this and incorporated the recent ASPEN journal guidelines (lines 220-257).

In reference to your comments regarding structural and histologic changes, we have included a section regarding hematologic/bone marrow changes (lines 581-596). In addition, though we addressed primarily the macro-changes in cardiac function, we added a section on myocardial fibrosis (lines 525-534). In addition we added a section on liver pathology (lines 466-477). Given this review is focused primarily on AN-R rather than AN-BP, we did not discuss the changes due to laxative and diuretic abuse with subsequent renal failure.

We clarified the role of functional GI complaints in AN. We have now included the Hata reference (lines 403-407, lines 433-440).

In reference to your suggestion RE: COVID and AN, though there has been increased prevalence on ED, we felt there is too little known at this point to include it in a review of known complications directly from AN. We did not feel that was within the scope of this review.

Round 2

Reviewer 2 Report

Thanks for this improved version of the manuscript.

I do agree that it is a comprehensive review of complications of AN, but not of new findings in AN. For the later to be true there are a bunch of imaging studies that needs to be discussed, likewise studies exploring other markers in blood than the neuropeptides and hormones mentioned here, e.g. inflammatory markers, markers of neuronal and glial loss. Another part that is missing is the link with E.Coli, production of ClpB peptide, and alphaMSH autoantibodies.

Since dieting is the most common trigger of AN, the statement below feels if not untrue at least incorrectly phrased.

"Interestingly, sociocultural factors, such as the pursuit of the thin-ideal and pressure to diet, have not been directly implicated in the pathogenesis of AN, as has been the case for bulimia nervosa and binge-eating disorder [19]."

In some cases you write HPA and in some HPAA. The former seems more correct.

r. 467. Serum methanol has been shown to be increased in AN. (Salehi et. al. 2021.), might be worth mentioning here.

Author Response

Since dieting is the most common trigger of AN, the statement below feels if not untrue at least incorrectly phrased.

"Interestingly, sociocultural factors, such as the pursuit of the thin-ideal and pressure to diet, have not been directly implicated in the pathogenesis of AN, as has been the case for bulimia nervosa and binge-eating disorder [19]."

Response: This statement has been revised. See lines 68-73.

In some cases you write HPA and in some HPAA. The former seems more correct.

Response: These are two different things, Hypothalamic Pituitary Axis and Hypothalamic Pituitary Adrenal Axis. Thus both are correct and abbreviated to reflect this.

r. 467. Serum methanol has been shown to be increased in AN. (Salehi et. al. 2021.), might be worth mentioning here.

Response: We did not feel this to be relevant to this particular manuscript.